# Anticipated stigma and associated factors among chronic illness patients in Amhara Region Referral Hospitals, Ethiopia: A multicenter cross-sectional study

Mohammed Hassen Salih[1]*, Hussen Mekonnen[2], Lema Derseh[3], Helena Lindgren[4], Kerstin Erlandsson[5]

1 University of Gondar, College of Medicine and Health Sciences, School of Nursing, Gondar, Ethiopia,
2 Addis Ababa University, School of Nursing and Midwifery, College of Nursing and Midwifery, Addis Ababa, Ethiopia, 3 University of Gondar, College of Medicine and Health Sciences, Institute of Public Health, Gondar, Ethiopia, 4 Department of Women's and Children's Health, Karolinska Institute, Solna, Sweden, 5 Department of Women and Children's Health, Karolinska Institute, Karolinska Institutet, Solna and School of Health and Welfare, Dalarna University, Falun, Sweden

☉ These authors contributed equally to this work.
* muhenet@gmail.com

**Data Availability Statement:** All relevant data are within the manuscript and its Supporting Information files. Also, our data is availed at a

## Abstract

### Background

Anticipated stigma related to chronic illness patients undermines diagnosis, treatment, and successful health outcomes. The study aimed to assess the magnitude and factors associated with anticipated stigma among patients with chronic illness attending follow-up clinics in Amhara Region Referral Hospitals, Ethiopia.

### Methods

A cross-sectional institution-based study was conducted in Amhara Region Referral Hospitals from 01 March to 15 April 2021. A simple random sampling technique was used to select the three Referral Hospitals in the region and study subjects. Data were collected using a pre-tested interview-based questionnaire. Data were entered and cleaned with Epi-Info version 6 and exported for analysis STATA version 14. Multiple linear regression was used to show the association between anticipated stigma and potential factors. Associations were measured using ß coefficients and were considered statistically significant if the p-value > 0.05.

### Results

A total of 779 patients were included for analysis with a response rate of 97%. Their mean (Standard deviation) of anticipated stigma was estimated at 1.86 and 0.5, respectively. After running an assumption test for multiple linear regression; educational status, cigarette smoking, psychological distress, medication adherence, alcohol consumption, and social part of the quality of life were statically significantly associated with anticipated stigma.

Mendley depository with V1, doi: 10.17632/m8y5cygdnm.1. However, sensitive identification was removed.

**Funding:** The author(s) received no specific funding for this work.

**Competing interests:** The authors have declared that no competing interests exist.

**Abbreviations:** AUDIT, Alcohol Use Disorder Identification Test; CIASS, Chronic Illness Anticipated Stigma Scale; DRH, Dessie Referral Hospital; FHRH, Felege Hiowt Referral Hospital; IQR, Inter Quartile Range; SD, Standard Deviation; UGRH, University Of Gondar Referral Hospital; USA, United States of America; WHOQOL, World Health Organization Quality Of Life.

## Conclusion and recommendation

The result showed a high level of anticipated stigma reported among the participants. Emphasizing improving their social part of the quality of life, avoiding risky behaviors like alcohol consumption and cigarette smoking, access to health education for chronically ill patients, integrating mental health in all types of chronic disease, and developing strategies and protocols which will help to improve patient medication adherence to their prescribed medication will be crucial. This can provide a foundation for government andnon-governmental organizations, and researchers implementing evidence-based interventions and strategies on chronic care to address factors related to anticipated stigma.

## Introduction

One of the effects of population ageing is an increased prevalence of chronic diseases, also referred to as non-communicable diseases (NCDs) [1]. Most scholars have used interchangeably the definition of chronic disease and chronic illness. A disease that usually lasts for 03 months or longer and may get worse over time, it can usually be controlled but not cured we call it a chronic disease. However; *Chronic illness* is the personal experience of living with the condition that often accompanies chronic disease [2,3]. Worldwide, over 35 million people die each year of chronic disease, eight out of ten in middle and low-income countries [4,5]. Chronic disease is also a primary contributor to annual health care costs for many countries around the globe [5,6]. In Africa, especially in Sub-Saharan Africa (SSA), the chronic disease burden is increasing [7–10]. Chronic disease-related stigma experienced and perceived by the patients is an era relatively new phenomenon that needs to be addressed by researchers and targeted in interventions. In the current pandemic of COVID-19, health-related stigma has increased [11]. This has led to a reduction in health-seeking at health institutions among chronic illness [12,13].

Stigma is a mark of social disgrace and is often associated with people deemed to have fallen outside of a socially-constructed norm. Stigmatizing is a process in which social meaning is attached to that individual due to his/her illness or disease [7]. The concept of stigma was first identified by Erving Goffman, who states that "spoiled identity" and social rejection for having a particular attribute [14], included loss of social status [15].

Since then, scholars have identified different types of stigma; enacted, anticipated and perceived stigmas are the most common [16,17]. For our study of chronic illness, we have decided to focus on the anticipated stigma. Anticipated stigma is an individual belief or perception that, because of their appearance or condition, prejudice, discrimination, and stereotyping will likely occur in the future in other words; it is an anticipation of an enacted stigma [18–22]. People living with chronic illness who anticipate stigma(anticipated stigma) are more likely to have experienced enacted stigma and to have internalized their experience of stigma in the past [23,24].

Patients with chronic illnesses see the illness itself as the source of their stigma. The illness can be the 'reason' for a 'diminished quality of life. Or it could be the 'problem' or 'factor' behind something else. It places them in conflict with workplace values such as productivity, and efficiency [25]. They may be perceived as unpredictable because they may experience periods of health despite their disability diagnosis [26].

Anticipated stigma among different chronic disease patients is evident and most common in resource-limited countries, like Ethiopia. Delivering high-quality care and managing the implications of chronic illness is one of the primary goals of the Ethiopian ministry of health. If the psychological components of chronic illness are not addressed, anticipated stigma and other conditions will affect the quality of care for chronic illness patients [27,28].

Despite the existing research on stigma and health, more work should be done on the extent to which people living with chronic illnesses anticipate stigma. Research has shown that anticipated stigma is attributable to different factors. Therefore, assessing the level and factors related to anticipated stigma among chronic illness patients will have tremendous output on how to enable the behavioral challenges presented by chronic illness patients with anticipated stigma to be addressed. Chronic care professionals including nurses need to be able to assess the magnitude of the anticipated stigma their patients might experience from friends and family members, work colleagues, and health care providers.

A considerable amount of stigma-related health research has been conducted in high-income countries [29–38]. As yet there is a paucity of studies related to anticipated stigma in low and middle-income countries. This study is the first to examine anticipated stigma amongst chronically ill patients in Ethiopia using validated and psychometrically-tested tools.

Currently, there are no published articles that show the magnitude of anticipated stigma among patients with chronic illness in Ethiopia. Therefore, there is a need to assess the level of anticipated stigma in the country and the factors which have contributed to them.

## Materials and methods

### Study area and period

A cross-sectional institution-based study was conducted in Amhara Region Referral Hospitals from 01 March to 15 April 2021. Amhara Region has located in the northwest part of the country and comprises the area traditionally occupied by ethnic Amharas, according to the last census taken in 2007 the Amhara region has a total population of approximately 17,221,976, of which 2,112,595 are based in urban and 15,109,381 in rural areas [39].

The region has 5 large referral hospitals to which local cases are routinely referred: Felege-Hiwot Referral Hospital (FHRH), Dessie Referral Hospital (DRH), Debre Birhan Referral Hospital (DBRH), Debre Markos Referral Hospital (DMRH), and University of Gondar Referral Hospital (UGRH). Three of these hospitals (Felege-Hiwot, Dessie, and the University of Gondar) were randomly selected to be part of this study.

### Population, sample size, and sampling procedure

The source for this study, and the study population itself, were the patients suffering from chronic illness who presented themselves for treatment at one of the three selected hospitals during the data collection period. Study participants had to be patients suffering from. hronic illness aged 18 or above who had been receiving follow-up treatment in a follow-up clinic for at least one year. fulfilled the cognitive criteria for mental illness [40] and were willing to volunteer their time for the study. Patients currently suffering from an active mental illness at the time of data collection were excluded from the study.

Because we were using a continuous outcome variable and for a single population the formula for determining sample size is $ni = \frac{((z_{\frac{\alpha}{2}}^{2})(S)^{2})2}{d^{2}}$. Where ni is the minimum sample size, S is the sample variance of anticipated stigma (0.75) determined from a previous pilot study [41], $Z_{\alpha/2}$ is a standardized normal distribution value at 95% confidence interval (CI)(1.96), and d is the margin of error of 0.05. By adding 10% for non-response and because we passed more

than one stage. By using a multiplication of 1.5 on the total number of possible patients we reached a figure of 802 as our maximum number of chronically ill patients eligible for inclusion in our study.

To determine inclusion from this group in our study, in each of the three referral hospitals a patient card identification number in the daily follow-up list was used as a sampling frame. Using proportional allocation, 183 patients from Dessie Referral Hospital, 269 patients from Felege Hiwot Referral Hospital, and 350 patients from the University of Gondar Referral Hospital were included in the study. A simple random sampling technique was used to get the proposed number of patients from each hospital. It took a maximum of two visits to each hospital has waited until we got the total participants in the study. Those participants who are not willing to participate were excluded and changed with volunteer ones to complete our desired sample size.

## Outcome variable and covariates

The outcome variable was the Amharic version of the chronic illness anticipated stigma scale (CIASS) adapted for use with patients in the Amhara region of Ethiopia The CIASS is a tool developed by Earnshaw and colleagues (2013) [42] used to measure anticipated stigma among study participants. It is comprised of twelve statements divided into three subscales, with four statements each seeking to evaluate the extent to which the patients anticipate stigma from friends and family members, work colleagues, and healthcare workers. Participants' responses were indicated on a Likert-type scale ranging from 1 (very unlikely) to 5 (very likely) [42]. Responses were then classified as "items" with the average anticipated stigma score was calculated by adding up all values of each of the items and dividing that by the total number of items [42]. The tool was administered in the Amharic and psychometrically tested and validated in the previous project [41].

Based on different literature and studies which were conducted before wenty-one variables were used as the independent (Covariate) variables. They were grouped into three sections; Sociodemographic (age, sex, marital status, residency, educational level, and occupation); Clinical (type of chronic illness, duration of chronic illness, type of medication, and comorbidity); and, personal and institutional (quality of life, psychological distress (K10), medication adherence, risky behaviors (alcohol, chat and/or cigarette), fear of contagion, regular follow-up, health information, clinical unit, and health institution).

Psychological distress was measured using the Kessler 10 distress scales [43], a valid and widely-used measure within research on Ethiopia [44–46]. The 2001 Victorian Population Health Survey cut-off scores were used for the regression analysis [47].

The World Health Organization Quality of Life (WHOQOL)-BREF measure was used to establish the quality of life. The English version of this 26-item instrument was translated into the Amharic Version with reliability analysis results showing Cronbach's alpha value of above 0.7 [48].

Medication adherence was measured by the Simplified Medication Adherence Scale (SMAS) consisting of six items, adopted from the Morisky Medication Adherence Scale (MMAS). This tool was also translated into Amharic by another author. A participant with scores of 12 then the patient was considered an 'adherent' to medication while those with 11 or less were considered 'non-adherent' to medication. Permission to use the Amharic version was secured before our data collection [49,50].

The degree of participation in risk behaviors such as alcohol/chat/cigarette smoking was measured. When measuring alcohol risky behavior we used the Alcohol Use Disorder Identification Test (AUDIT) tool that has been validated for use with the Amharic-speaking

population, of Ethiopia [51,52]. When measuring cigarette smoking, we adopted the WHO guidelines in a questionnaire and then placed the patient into the categories of either non-smokers, former smokers, occasional smokers, or daily smokers. Non-smokers were defined as those who had never smoked or who, at the time of the study, had been smoking for less than one month. Former smokers (ex-smokers) were defined as those who had previously had a daily smoking habit for a continuous period of at least six months but had given up smoking at least one month before the completion of the questionnaire. Occasional smokers were defined as those who did not smoke daily. Daily smokers were defined as those who smoked at least one cigarette per day for at least one month before completing the questionnaire [53].

## Data collection procedure

Data was collected by 12 nurses, each with a BSC degree. The data collection process was supervised by 03 health professionals, each with MSc-holder. Three-day training was delivered for the data collectors and supervisors by the principal investigators. The principal investigators have coordinated the data collection process with regular input from the data collectors and supervisors. Before the actual data collection process, a pilot test comprising 10% of the total sample was carried out at the University of Gondar Referral hospital by careful consideration for not included in the current study. This showed inter-rater reliability of 81% for intra-class correlation. The internal consistency of the Cronbach alpha was 80.1%.

## Data analysis

Every completed response to the CIASS measuring tool was checked visually for completeness before the data were entered into the computer. The data were entered into Epi Info v.7 before being transported to and analyzed by using STATA 14 and IBM SPSS V-26. Descriptive statistical tests like frequencies, mean, standard deviation, and unpaired sample two-way T-tests, were performed.

Before moving on to the linear regression model assumption, a model fitness test was carried out to determine if there was a clustering effect at each hospital level and/or clinical unit. These tests showed that the clustering effect was below the recommended levels. Therefore, there was no clustering effect and a linear model could be used. After checking the assumption in the linear model and model fitness test, a regression analysis was performed. Factors associated with anticipated stigma were considered at a p-value of 0.05 and below for a 95% confidence interval.

## Ethical issues

Ethical approval for this study was received from the University of Gondar ethical review board (IRB) on September 17[th], 2020 (V/P/RCS/05/89/2020). Written consent was obtained from the relevant authority within each of the participating hospitals. The written signature of each patient was obtained before their data was collected.

During the data collection period, the principal investigator and data collectors were ensured confidentiality and privacy of the information they had collected. Information sheets and consent forms were prepared and circulated before the collection of data. Patients who reported a high level of stigma and indicated a desire for further clinical and psychological management were put in touch with a clinical psychologist for counseling and the opportunity to identify the source of their stigma. The finding from this study will be disseminated to health institutions, policy-makers, and other c interested parties through academic articles, policy statements, and other printed and published texts.

## Results

### Sociodemographic characteristics

A total of 779 follow-up patients seeking follow-up treatment for chronic illness were included in this study, with a response rate of 97%. More than half (55%) of them were female. The mean (± SD) age of the patients was 42.3 (± 13.3) and just over half (51%) of these fall were in the age range of 31–50 years. A large number (74%) resided in urban areas.

Almost all patients (97%) were ethnic Amhara's and a significant proportion (78%) were orthodox Christians. Concerning their educational status, more than one quarter (29%) of them had above primary educational level. Six out of ten (58%) were married (Table 1).

### Personal and institutional characteristics

A significant proportion (81.9%) of the patients had acquired information related to their health in different media sources. Six out of ten (63.41%) had heard health information on the radio. A significant proportion (81%) reported they had fear of contagion and just over four-fifths (84.47%) had regular follow-up at their health institution. A small minority (9%) reported a history of cigarette smoking and less than a fifth (16%) claimed to chew chat. Those who chewed chat had a median of 50gm and an Inter Quartile Range (IQR) of 25 gm chat chewing per day.

The median and IQR for medication adherence were 12 and 2 respectively, with more than half (56%) claiming adherence to their prescribed drugs. The AUDIT tool that measured alcohol intake showed a median of 0 and IQR of 4. Four-fifths (81%) of patients were classified as having no alcohol use.

On Kessler psychological distress scale the median and IQR were 21 and 12 respectively; More than two-fifths (43%) of patients were classified as "likely to be well" on the psychological distress signs scale. The highest level of mean and SD (2.09 & 0.45) of anticipated stigma was reported by patients with severe mental disorders.

The mean and SD of Physical QOL were 51.5, and 17.4, Psychological QOL 56.7, and 18.2, Social QOL 54.2, and 21.3, and Environmental QOL 53, and 17.3 respectively (Table 2).

### Clinical characteristics

The highest level of the mean (1.96) was reported among diagnoses of mental illness and they covered one quarter (26%) of the other's diagnosis. The median and IQR length of since first 6 and 7 years respectively. Approximately half (49%) of the patients received to know their diagnosis 1–5 years ago and the anticipated stigma mean and standard deviation for this group were 1.88 and 0.5 respectively. Almost a quarter (24%) had received a diagnosis (more than one) for an additional condition; the highest mean level (1.92) was reported among this group. (Table 3).

### Levels and factors associated with anticipated stigma

Based on the total number of 779 patients in this study, the mean and SD of anticipated stigma among chronic illness patients were 1.86 and 0.5, respectively. A relatively higher mean and SD of anticipated stigma were reported by colleagues working in a similar area with figures of 2.1 and 0.8, respectively (Fig 1).

After checking the assumptions for multiple linear regressions, six out of twenty independent variables were found to be associated with anticipated stigma. The R-squared was estimated at 0.2331, meaning that approximately 23% of the variability of the anticipated stigma levels was accounted for by the variables in the model, even after taking into account the number of predictor variables in the model (P-value < 0.001). Educational status, cigarette smoking

**Table 1. Sociodemographic characteristics of patients with chronic illness in three Amhara region Referral hospitals, Northwest Ethiopia (n = 779).**

| Variables | | Number (%) | Anticipated stigma Mean (SD) | T-test/ ANOVA (p-value) |
|---|---|---|---|---|
| **Sex** | Male | 354 (45.44) | 1.90 (0.52) | 0.1831 |
| | Female | 425 (54.56) | 1.85 (0.50) | |
| **Age groups (Years) (Mean 42.3, and SD = 3.3)** | | | | |
| | 30 or less | 183 (23.49) | 1.91 (0.52) | 0.3550 |
| | 31–40 | 198 (25.42) | 1.85 (0.53) | |
| | 41–50 | 199 (25.55) | 1.86 (0.51) | |
| | 51–60 | 131 (16.82) | 1.92 (0.50) | |
| | 61 or more | 68 (8.73) | 1.79 (0.38) | |
| **Residency** | Urban | 576 (73.94) | 1.87 (0.51) | 0.6778 |
| | Rural | 203 (26.06) | 2.11 (0.50) | |
| **Ethnicity** | Amhara | 758 (97.30) | 1.89 (0.50) | 0.2039 |
| | Oromo | 08 (1.03) | 2.19 (0.60) | |
| | Others | 13 (1.67) | 1.83 (0.60) | |
| **Religion** | Orthodox | 609 (78.18) | 1.86 (0.51) | |
| | Muslim | 164 (21.05) | 1.94 (0.50) | 0.1913 |
| | Others | 06 (0.77) | 1.79 (0.48) | |
| **Educational status** | Not read and write | 179 (22.98) | 1.80 (0.45) | |
| | Read and write | 129 (16.56) | 1.85 (0.45) | |
| | Primary school | 128 (16.43) | 1.95 (0.49) | 0.1238 |
| | Secondary school | 119 (15.28) | 1.89 (0.55) | |
| | Above secondary school | 224 (28.75) | 1.90 (0.55) | |
| **Marital status** | Single | 172 (22.08) | 1.91 (0.54) | |
| | Married | 450 (57.77) | 1.85 (0.49) | 0.2168 |
| | Separated/widowed | 78 (10.01) | 1.88 (0.48) | |
| | Divorced | 79 (10.14) | 1.96 (0.55) | |
| **Occupation** | Governmental | 152 (19.51) | 1.88 (0.54) | |
| | NGO | 45 (5.78) | 1.93 (0.42) | 0.0628 |
| | Self | 369 (47.37) | 1.84 (0.48) | |
| | Others | 213 (27.34) | 1.91 (0.54) | |

Key:—SD = Standard deviation, N/A = not applicable, NGO = Non-Governmental Organizations.

and alcohol consumption, psychological distress, medication adherence, and the social dimension of quality of life were associated with anticipated stigma

Educational status, cigarette smoking and alcohol drinking of personal behaviors, psychological distress, medication adherence, and social part of the quality of life were associated with anticipated stigma.

The value of the anticipated stigma score was 0.137 higher for patients with a primary educational level than those who do not write and read ($p < 0.05$). Patients Above secondary school had scores that were 0.13 higher than those who could not read and write ($p < 0.05$) when all other variables in the model were held constant.

In the personal risky behavioral factors; The value of the anticipated stigma score was 0.29 higher for those who had a history of smoking than for those who had never smoked ($p < 0.001$), assuming the other variables were kept constant. And Patients who did not drink alcohol had on average an anticipated stigma score that was 0.10 scores lower than those who drink alcohol ($p < 0.05$) when all other variables were kept constant.

**Table 2. Personal and institutional characteristics of patients with chronic illness in three Amhara region Referral Hospitals, Northwest Ethiopia (n = 779).**

| Variables | | Number (%) | Anticipated stigma Mean (SD) | T-test (p-value) |
|---|---|---|---|---|
| Have information on health-related | No | 141 (18.10) | 1.92 (0.52) | 0.2526 |
| | Yes | 638 (81.90) | 1.86 (0.50) | |
| Radio | No | 285 (36.59) | 1.83 (0.51) | 0.0537 |
| | Yes | 494 (63.41) | 1.90 (0.50) | |
| Magazine | No | 665 (85.37) | 1.86 (0.51) | 0.1554 |
| | Yes | 114 (14.63) | 1.94 (0.51) | |
| Television | No | 322 (41.34) | 1.88 (0.50) | 0.9155 |
| | Yes | 457 (58.66) | 1.87 (0.51) | |
| Fear of contagion | No | 150 (19.26) | 1.94 (0.56) | 0.0970 |
| | Yes | 629 (80.74) | 1.86 (0.49) | |
| Regular follow-up | No | 121 (15.53) | 1.98 (0.59) | 0.0390 |
| | Yes | 658 (84.47) | 1.86 (0.49) | |
| Cigarette smoking | No | 709 (91.01) | 1.84 (0.49) | **0.0000** |
| | Yes | 70 (8.99) | 2.24 (0.54) | |
| Chat chewing | Never | 652 (83.70) | 1.84 (0.50) | **0.0001** |
| | Yes | 127 (16.30) | 2.05 (0.53) | |
| Currently chewing chat | Never | 727 (93.32) | 1.86 (0.50) | **0.0063** |
| | Yes | 52 (6.68) | 2.08 (0.54) | |
| Alcohol intake | Yes | 151 (19.38) | 1.99 (0.55) | **0.0042** |
| | Not alcohol | 628 (80.62) | 1.85 (0.49) | |
| Medication adherence | Adhered | 434 (55.71) | 1.76 (0.48) | **0.0000** |
| | Not adhered | 345 (44.29) | 2.02 (0.51) | |
| Psychological distress likely to | Be well in distress | 337 (43.26) | 1.69 (0.51) | **0.0000** |
| | Have mild disorder | 174 (22.34 | 1.96 (0.49) | |
| | Have a moderate disorder | 154 (19.77) | 2.01 (0.44) | |
| | Have a severe disorder | 114 (14.63) | 2.09 (0.45) | |

Key:—SD = Standard Deviation.

**Table 3. Clinical characteristics of patients with chronic illness in three Amhara region Referral Hospitals, Northwest Ethiopia (n = 779).**

| Variables | | Number (%) | Anticipated stigma Mean (SD) | T–test (p-value)/(ANoVA) |
|---|---|---|---|---|
| Type of chronic illness | Hypertension | 100 (12.84) | 1.90 (0.47) | 0.1563 |
| | Dm | 162 (20.80) | 1.85 (0.46) | |
| | Cancer | 78 (10.01) | 1.75 (0.51) | |
| | Hiv/aids | 161 (20.67) | 1.83 (0.58) | |
| | Mentally ill | 202 (25.93) | 1.96 (0.49) | |
| | Others* | 76 (9.76) | 1.88 (0.50) | |
| Comorbidity | No | 594 (76.25) | 1.86 (0.51) | 0.1642 |
| | Yes | 185 (23.75) | 1.92 (0.50) | |
| Year of diagnosis (years) | ≤ 05 | 384 (49.29) | 1.88 (0.50) | 0.0623 |
| | 5–10 | 262 (33.63) | 1.89 (0.52) | |
| | ≥10 | 133 (17.07) | 1.82 (0.50) | |

N.B: DM = Diabetes Mellitus.

* Includes Cardiac, Neurologic, Hepatitis, and Arthritis.

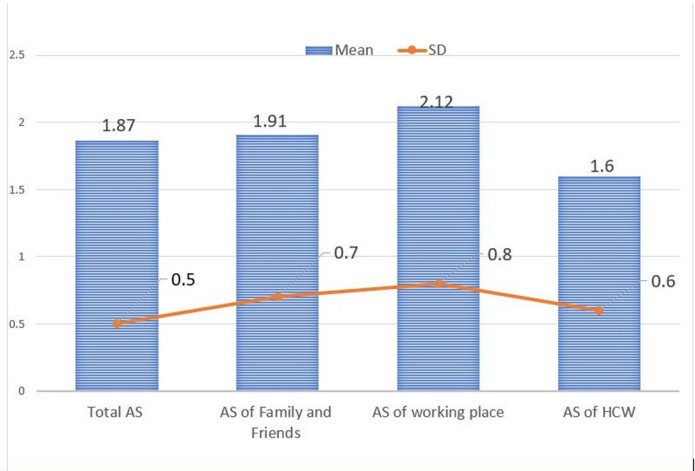

Keys – AS = Anticipated Stigma, HCW = Health Care Workers, SD = Standard Deviation

**Fig 1. Anticipated stigma level.**

On Kessler psychological distress scale (K10), anticipated stigma was found to be 0.34 points higher for patients claiming likely to have a severe disorder than those claiming likely to be well (p < 0.001), 0.27 points higher for patients claiming likely to have a moderate disorder and 0.21 points higher for patients claiming to have a mild disorder, assuming that all other variables in the model are held constant.

For every unit increase in the social quality of life experienced by the patient in this study, there was on average decrease in the value of the anticipated stigma score by 0.005 (p < 0.001), considering all other explanatory variables in the model are kept constant.

Patients who had not adhered to their medication regimes scored 0.18 higher on the anticipated stigma scale than those who did adhere to their prescribed medication (p < 0.001), assuming that all other variables in the model were held constant (Table 4).

## Discussion

Anticipated Stigma was a major social determinant of health that can lead to morbidity, mortality, and health disparities among chronic illness patients. This study is, to our knowledge, the first to have used the Amharic version of the anticipated stigma scale for patients with chronic illness, an instrument known to have good reliability and validity for evaluating the level of anticipated stigma among chronic illness patients in Ethiopia.

The mean and SD of anticipated stigma among chronic illness patients were 1.86 and 0.5, respectively. Our results were almost similar to those found in a study conducted among Chinese stroke patients and lower than those presented in a study of neurological and chronic obstructive pulmonary disease (COPD) patients in the USA [29,54]. The possible justification for the discrepancies might be that our study incorporated patients experiencing all types of chronic illnesses not simply one distinct category and the sample representativeness of the USA study. This may exaggerate the level of anticipated stigma. Nurses still need to be vigilant and look to address not only the pathophysiological but also their psychosocial problems contributing to stigma for patients with chronic illness.

The value of the anticipated stigma score was 0.137 higher for patients with a primary educational level than those who do not write and read (p < 0.05). Patients Above secondary school had scores that were 0.13 higher than those who could not read and write (p <0.05) when all other variables in the model were held constant.

**Table 4. Factors associated with anticipated stigma among patients with chronic illness in three Amhara Region Referral Hospitals, Northwest Ethiopia (n = 779).**

| Source | SS | Df | MS | Number of obs = 779 |
|---|---|---|---|---|
| Model | 46.6174732 | 11 | 4.23795211 | **F(11, 767)** = 21.20 |
| Residual | 153.345888 | 767 | .19992945 | **Prob > F** = 0.0000 R-squared = 0.2332 |
| Total | 199.963361 | 778 | .257022315 | Adj R-Squared = 0.2221 Root MSE = 0.44713 |

| Anticipated stigma | Coefficient (ß) | P - value | [95% Conf. Interval] | |
|---|---|---|---|---|
| **Educational status** (Reference = Not write and read) | | | | |
| Read and write | 0.027 | 0.608 | -0.075 | 0.128 |
| Primary education | 0.137 | **0.009** | 0.034 | 0.240 |
| Secondary | 0.086 | 0.106 | -0.018 | 0.190 |
| Above Secondary school | 0.125 | **0.006** | 0.036 | 0.214 |
| **Cigarette smoking** (Reference = never) | | | | |
| Yes | 0.291 | **0.000** | 0.179 | 0.405 |
| **K10** (Reference = likely to be well in psychological distress) | | | | |
| Likely to have a mild disorder | 0.207 | **0.000** | 0.124 | 0.290 |
| Likely to have a moderate disorder | 0.268 | **0.000** | 0.181 | 0.355 |
| Likely to have a severe disorder | 0.344 | **0.000** | 0.247 | 0.441 |
| **Social QOL** | -0.005 | **0.000** | -0.006 | -0.003 |
| **Alcohol drinking** (Reference = alcohol drinker) | | | | |
| Not-Alcohol drinker | -0.098 | **0.017** | -0.179 | -0.018 |
| **Medication adherence** (Reference = medication adhered to) | | | | |
| Medication not adhered | 0.182 | **0.000** | 0.117 | 0.247 |
| **Constant** | **1.872** | **0.000** | **1.737** | **2.008** |

Keys:—SS = sum square, df = degree of freedom, MS = Mean Square, K10 = Kessler psychological distress scale, QOL = Quality of life.

This result is supported by a study conducted in Asia and the World Health Mental Survey [31,33]. Education increases the level of anticipated stigma [55]. A proper educational strategy for all healthcare professionals including nurses needs to be developed that will enable chronic disease patients to talk about stigma and discrimination.

The value of the anticipated stigma score was 0.29 higher for those who had a history of smoking than for those who had never smoked (p < 0.001), assuming the other variables were kept constant. Risky behaviors such as smoking have been documented as one of the factors increasing anticipated stigma in Asia and one among stroke-related patients in Africa [31,56]. Cigarette smoking can lead to lower social interaction and is possibly related to other risky personal behaviors. This can lead to chronically ill patients being reluctant to disclose their status to others, which in turn can lead to increased anticipated stigma towards them. Nurses working in chronic care clinics and other rehabilitative centers always need to assess patients displaying risk behaviors that may worsen their disease status and, by extension, their level of stigma. Patients may need to be referred to other specialist treatment clinics, such as those for nicotine addiction.

When the results of the Kessler psychological distress scale (K10) were examined, and assuming that all other variables in the model were held constant, anticipated stigma was found to be 0.34 points higher for patients claiming likely to have a severe disorder than those claiming likely to be well (p < 0.001), 0.27 points higher for patients claiming likely to have a moderate disorder and 0.21 points higher for patients claiming to have a mild disorder. This finding is also supported elsewhere in the published literature. Studies have shown that anticipated stigma increased in direct proportion to the severity of mental health distress signs

[22,37,57,58]. Different levels of stress affect patients of any age or socioeconomic status suffering from chronic illness. For this reason, chronic care mental health nurses may need to work with patients suffering from many different types of chronic diseases [59].

For every unit increase in the social quality of life experienced by the patient in this study, there was on average decrease in the value of the anticipated stigma score by 0.005 (p < 0.001), considering all other explanatory variables in the model are kept constant. This finding clearly showed that the more quality of life improves, the less the anticipated stigma experienced. This finding is also supported by the research conducted on chronic illness patients in different parts of the world [29,31,56,57,60–64]. These studies show that Quality of Life (QoL) is a perception of a patient has a relative social position. Delivering care that meets the patient's unique needs (social quality of life) requires nurses and healthcare providers to have a sensitive and nuanced understanding of their quality of life [65].

Patients who did not drink alcohol had on average an anticipated stigma score that was 0.10 scores lower than those who drink alcohol (p < 0.05) when all other variables were kept constant. This finding is also supported by existing research, in this case, a study conducted among patients who developed cirrhosis of the liver [57]. Patients suffering from long-term alcohol abuse may be vulnerable to an increased risk of other chronic diseases and complications. This may lengthen their recovery times which can, in turn, trigger an increase in the anticipated stigma [66,67].

Patients who had not adhered to their medication regimes scored 0.18 higher on the anticipated stigma scale than those who did adhere to their prescribed medication (p < 0.001), assuming that all other variables in the model were held constant. Medication adherence is related to patient improvement and reduced stigma in many chronic illness studies, including several conducted in Cambodia, Myanmar, Vietnam [31], one carried out among West African Stroke patients [56], and one Nigerian study focusing on psychiatric patients [40]. The reason for this positive correlation could be that adherence to the prescribed medication regime can be attributed to growing knowledge and understanding of the progress and development of the chronic disease, and a greater understanding of how others perceive it [68]. Medication adherence was the behavioral tool used in these studies to improve patient quality of care and control disease complications. Improving medication adherence was critically important, but it was often difficult for patients to sustain this objective. Nurses are often the last line of defense when encouraging patients to takin their medication. Health care professionals working in chronic care, therefore, need to develop strategies and protocols that can increase patient adherence for their prescribed medication, like adherence triad.

This study shows the possible consequences of anticipated stigma on the health and psychological behavior of people living with chronic diseases. Understanding the factors that contribute to anticipated stigma can help to guide interventions that aim to reduce the stigma attached to chronic illness.

This research has some limitations. There may be a social desirability bias because the topic is relatively sensitive and not often discussed. The participants may have chosen only socially acceptable answers, and we cannot rule out the cause-effect relationship created by the study's design.

This study is the first to examine such population groups within a developing country using a specifically adapted version of a widely studied and reliable tool for analyzing anticipated stigma with the full range of chronic illnesses where the strength of the study.

In a conclusion, this study shows a high level of anticipated stigma reported by chronic illness patients in Northwestern Ethiopia. Educational status, cigarette smoking, psychological distress, the social side of the quality of life, alcohol drinking, and adherence to their prescribed medication were all factors related to anticipated stigma levels.

## Supporting information

**S1 File.**
(XLS)

**S1 Questionnaire.**
(DOCX)

## Acknowledgments

We would like to thank all of the patients, data collectors, and facilitators (Mr. Alebachew G, Dr. Gashaw A, Dr. Teshome M, and Sr. Belaynesh T.) who participated in this study for their support throughout the data collection period. We also want to extend our thanks to Professor Valerie Earnshaw for her permission to use the chronic illness anticipated stigma scale. Last but not least, would like to appreciate the effort in language editing by Dr. Janice Holms.

## Author Contributions

**Conceptualization:** Mohammed Hassen Salih, Hussen Mekonnen, Lema Derseh, Helena Lindgren.

**Data curation:** Mohammed Hassen Salih, Helena Lindgren, Kerstin Erlandsson.

**Formal analysis:** Mohammed Hassen Salih, Lema Derseh, Kerstin Erlandsson.

**Funding acquisition:** Hussen Mekonnen.

**Investigation:** Mohammed Hassen Salih, Lema Derseh, Helena Lindgren, Kerstin Erlandsson.

**Methodology:** Mohammed Hassen Salih, Hussen Mekonnen, Lema Derseh, Helena Lindgren, Kerstin Erlandsson.

**Project administration:** Hussen Mekonnen, Helena Lindgren, Kerstin Erlandsson.

**Resources:** Mohammed Hassen Salih, Helena Lindgren.

**Software:** Mohammed Hassen Salih, Lema Derseh, Kerstin Erlandsson.

**Supervision:** Hussen Mekonnen, Lema Derseh, Kerstin Erlandsson.

**Validation:** Hussen Mekonnen, Lema Derseh, Helena Lindgren, Kerstin Erlandsson.

**Visualization:** Mohammed Hassen Salih, Hussen Mekonnen, Lema Derseh, Helena Lindgren, Kerstin Erlandsson.

**Writing – original draft:** Mohammed Hassen Salih, Lema Derseh, Kerstin Erlandsson.

**Writing – review & editing:** Mohammed Hassen Salih, Kerstin Erlandsson.

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
