## [Decision Letter · Decision Letter 0]

28 Jun 2022

PONE-D-22-00465Anticipated stigma and associated factors among chronic illness patients in Amhara Region Referral Hospitals, Ethiopia: a multi-center cross-sectional study.PLOS ONE

Dear Dr. Mohammed,

Thank you for submitting your manuscript to PLOS ONE. After careful consideration, we feel that it has merit but does not fully meet PLOS ONE’s publication criteria as it currently stands. Therefore, we invite you to submit a revised version of the manuscript that addresses the points raised during the review process.

**Comments from academic editor:**

Abstract-conclusion/recommendation: the recommendation reads too general. It would be better if some specific recommendations can be made based on the study findings (the identified associated factors).Introduction: “In the current pandemic COVID-19, health-related stigma has increased”, please add reference(s) for this sentence.Introduction: “scholars have identified different types of stigma; enacted, anticipated and perceived stigmas are the most common.” Reference is needed.Introduction: “Anticipated stigma is an individual belief or perception that…..” Reference is needed.Materials and Methods- Study area and period: in this section, the monthly number of follow-up patients in each hospital was reported, please specify if these were patients with chronic diseases. Otherwise, please consider deleting this information.What is the sampling method for participant recruitment? Please specify.Sample size calculation: “S is the sample variance of anticipated stigma (0.75) determined from a previous pilot study”, please add the reference for the mentioned previous pilot study.“Twenty-one variables were used as the independent (Covariate) variables.” How did you determine those 21 variables? Based on previous studies or any theoretic framework, please specify.Data analysis: Were all the 21 independent variables directly included in the regression analysis? Any univariate analysis and/or correlation analysis was conducted prior to the regression analysis to identify the variables with statistical significance that can be included in the regression analysis?

     10. Results and discussion: please separate the results and discussion into two sections.      

**Reviewers' comments:**

Reviewer #1: Manuscript comments

When I review this paper, I have found some strength and some comments to be improved. The paper is well stated and the title is very crucial, very important and timely searched particularly in developing country. The model selection for analysis was appropriate for such type of continuous outcome variable. The result and discussion were described properly. However, I have some comments to be improved the manuscript which is listed below.

1. In abstract: methods part better to put first “A simple random sampling technique was used to select the three Referral Hospitals in the region and study subjects.” then “Data were collected using a pre-tested interview based questionnaire.”

2. In introduction: better to incorporate about definition of chronic illness and put the definition of stigma (from line 68-72) is in the first paragraph.

3. In materials and methods part: The information which is written from line 120 to 122 should be updated. In data collection procedure part in line 203 (each with an MSc –holder or higher) needs edition because the data was already collected. From where the pilot studies was done? Should be mention.

We look forward to receiving your revised manuscript.

Kind regards,

Alison Wang

Academic Editor

PLOS ONE
---

## [Author Response · Author response to Decision Letter 0]

12 Jul 2022

Dear reviewer/professor, I thank you for the critical comments you gave me. All the comments and suggestions were excellent and scientific view. It increased our paper quality.

Dear Professor, we thank you for your valuable time and here we tried to respond line by line to each comment and suggestion.

When I review this paper, I found some strengths and some comments to be improved. The paper is well stated and the title is very crucial, very important, and timely searched particularly in a developing country. The model selection for analysis was appropriate for such type of continuous outcome variable. The result and discussion were described properly. However, I have some comments to be improved on the manuscript which are listed below.

Response – thank you professor for your excellent view of our paper and the suggestions.

1. In the abstract: methods part better to put first “A simple random sampling technique was used to select the three Referral Hospitals in the region and study subjects.” then “Data were collected using a pre-tested interview-based questionnaire.”

Response – Thank you professor and we correct it based on your excellent view.

2. In the introduction: better to incorporate about the definition of chronic illness and put the definition of stigma (from line 68-72) is in the first paragraph.

Response – yes professor, we incorporated your suggestion in the first paragraph. Thank you.

3. In materials and methods part: The information which is written from line 120 to 122 should be updated. In data collection procedure part in line 203 (each with an MSc –holder or higher) needs edition because the data was already collected. From where the pilot studies was done? Should be mention

Response- Dear Professor, I thank you for your critical review and comments. When we come to the population number of the Amhara Region we tried to get valuable data but not as much as the Ethiopian Statistical Agency. So, we agreed to use this one. 

In the Data collection procedure we correct based on your comment. Thank you so much

Dear journal academic editor 

Thank you for your important and scientific-based comments raised for my manuscript. Here I tried to respond to each comment line by line. Thank you.

1. Abstract-conclusion/recommendation: the recommendation reads too general. It would be better if some specific recommendations can be made based on the study findings (the identified associated factors).

Response - Dear professor, thank you for the comment. Incorporated the suggested point in the abstract.

2. Introduction: “In the current pandemic COVID-19, health-related stigma has increased”, please add a reference(s) for this sentence.

Response - I thank you and attached the reference in the specified sentence. 

3. Introduction: “scholars have identified different types of stigma; enacted, anticipated and perceived stigmas are the most common.” Reference is needed.

Response - I thank you and attached the reference in the specified sentence.

4. Introduction: “Anticipated stigma is an individual belief or perception that …....” Reference is needed.

Response - Thank you and revised the incorporated the reference.

5. Materials and Methods- Study area and period: in this section, the monthly number of follow-up patients in each hospital was reported, please specify if these were patients with chronic diseases. Otherwise, please consider deleting this information.

Response - Delete it and accept your suggestion.

6. What is the sampling method for participant recruitment? Please specify.

Response - Thank you, professor, to reach the study subject we used a simple random sampling method and it was stated line………. 

7. Sample size calculation: “S is the sample variance of anticipated stigma (0.75) determined from a previous pilot study”, please add the reference for the mentioned previous pilot study.

Response - I thank you, professor, for our “S” for the sample size taken from the previous project entitled Translation and psychometric evaluation of chronic illness anticipated stigma scale (CIASS) among patients in Ethiopia. Which was published in the same journal. We attach the reference based on the comment.

8. “Twenty-one variables were used as the independent (Covariate) variables.” How did you determine those 21 variables? Based on previous studies or any theoretic framework, please specify.

Response - Dear Professor, thank you and we select those variables from previous papers and different literature. We accept your comment and add a sentence to clearly stated how 21 variables were selected.

9. Data analysis: Were all the 21 independent variables directly included in the regression analysis? Any univariate analysis and/or correlation analysis conducted before the regression analysis to identify the variables with statistical significance that can be included in the regression analysis?

Response - Before we directly going to analyze our independent variables against the dependent variable, we did the basic assumption related to linear model regression. After the assumption was fulfilled, we directly included all those variables which were full filled the assumption. For our discussion section, we only discussed those variables that have a real association with the outcome variable. Thank you for your important insight and gave a much time to increase the validity of my manuscript. 

10. Results and discussion: please separate the results and discussion into two sections. 

Thank you and I was trying to decrease the volume of our paper and make it short. But when we see your comment it was important to separate the result and discussion section. Accept and separated into result and discussion sections. 

Response - Thank you for your time and have a good day.

---

## [Decision Letter · Decision Letter 1]

15 Aug 2022

Anticipated stigma and associated factors among chronic illness patients in Amhara Region Referral Hospitals, Ethiopia: a multi-center cross-sectional study.

PONE-D-22-00465R1

Dear Dr. Mohammed Hassen,

We’re pleased to inform you that your manuscript has been judged scientifically suitable for publication and will be formally accepted for publication once it meets all outstanding technical requirements.

Kind regards,

Alison Wang

Academic Editor

PLOS ONE

---

## [Editor Report · Acceptance letter]

19 Aug 2022

PONE-D-22-00465R1 

Anticipated stigma and associated factors among chronic illness patients in Amhara Region Referral Hospitals, Ethiopia: a multicenter cross-sectional study. 

Dear Dr. Salih:

I'm pleased to inform you that your manuscript has been deemed suitable for publication in PLOS ONE. Congratulations! Your manuscript is now with our production department. 

Kind regards, 

on behalf of

Dr. Tao (Alison) Wang 

Academic Editor

PLOS ONE